# Corn Cob Char as Catalyst Support for Developing Carbon Nanotubes from Waste Polypropylene Plastics: Comparison of Activation Techniques

**DOI:** 10.3390/polym14142898

**Published:** 2022-07-16

**Authors:** Helen U. Modekwe, Kapil Moothi, Michael O. Daramola, Messai A. Mamo

**Affiliations:** 1Department of Chemical Engineering, Faculty of Engineering and the Built Environment, University of Johannesburg, Doornfontein Campus, P.O. Box 17011, Johannesburg 2028, South Africa; uche_lyne2@yahoo.com (H.U.M.); kmoothi@uj.ac.za (K.M.); 2Department of Chemical Engineering, Faculty of Engineering, Built Environment and Information Technology, University of Pretoria, Private Bag X20 Hatfield, Pretoria 0028, South Africa; michael.daramola@up.ac.za; 3Research Centre for Synthesis and Catalysis, Department of Chemical Science, Faculty of Science, University of Johannesburg, Doornfontein Campus, P.O. Box 17011, Johannesburg 2028, South Africa

**Keywords:** waste corn cob, char activation, catalyst support, polypropylene, waste plastics, carbon nanotubes

## Abstract

The future and continuity of nanomaterials are heavily dependent on their availability and affordability. This could be achieved when cheap materials are actively employed as starting materials for nanomaterials synthesis. In this study, waste corn cob char was used as support during the preparation of the NiMo catalyst, and the effect of different char-activating techniques on the microstructure, yield and quality of carbon nanotubes (CNTs) obtained from waste polypropylene (PP) plastics using the chemical vapor deposition (CVD) technique was investigated. Properties of the catalysts and obtained nanomaterials were evaluated by XRD, SEM, N_2_ physisorption experiment, FTIR, Raman spectroscopy and TEM. Results showed improved surface properties of the NiMo catalyst supported on chemically (NiMo/AC_X_) and physically activated char (NiMo/AC_T_) compared to the NiMo catalyst supported on non-activated char (NiMo/AC_0_). High-quality CNTs were deposited over NiMo/AC_T_ compared to NiMo/AC_X_ and NiMo/AC_0_. It was also observed that different activation methods resulted in the formation of CNTs of different microstructures and yield. Optimum yield (470.0 mg CNTs/g catalyst) was obtained with NiMo/AC_0_, while NiMo/AC_T_ gave the least product yield (70.0 mg CNTs/g catalyst) of the as-produced nanomaterials. Based on the results of the analysis, it was concluded that utilizing a cheap pyrogenic product of waste corn cob as a catalyst support in a bimetallic NiMo catalyst could offer a promising approach to mass producing CNTs and as a low-cost alternative in CNTs production from waste plastics.

## 1. Introduction

The growing market for CNTs has expanded in recent years due to CNTs’ vast and innovative applications in energy storage (fuel cells. lithium batteries), hydrogen storage, sensors, nanoelectronics and nanoprobes (scanning probe microscope), field emitters, nanocomposites, tissue engineering and medicine (in drug delivery), gas and CO_2_ absorption, wastewater purifications, etc. [1,2,3,4,5]. CNTs are originally produced from high demand carbonaceous and low molecular weight gases such as methane, acetylene, ethylene, etc., resulting in relatively high prices of CNTs reaching up to USD 2000 and 100 per kilogram for SWCNTs and MWCNTs, respectively [6]. The high cost of CNTs and CNTs products impedes the exploration of other enormous potential applications which place cumulative demand on ways to reduce CNTs’ production costs and of course augment its affordability. In achieving this, other carbonaceous sources have been widely investigated such as coal [7], palm kernel shell [8], waste tires [9], waste plastics [10,11,12,13,14,15], etc. The extensive growth in human population and industrialization has resulted in a tremendous increase in household and industrial demand and application of plastic materials such as polypropylene, polyethylene (both low and high-density), polystyrene, etc. Since these materials are relatively non-biodegradable, they are generally discarded in landfills. Consequently, the volume of available landfill spaces is diminishing, and obtaining value-added products from waste plastics is pertinent in reducing the environmental and economic impact of waste plastics as well as reducing the cost of carbon nanotubes. 

The chemical vapor deposition (CVD) technique is considered the most compelling method for the large-scale production of CNTs of various morphologies and diameters. CNTs produced through CVD are achieved in the presence of carbonaceous feedstock and catalysts, amongst others. Nickel-based catalysts are widely studied due to their low cost and high catalytic activity as a result of their carbon solubility property [16]; they are usually supported on materials such as MgO, Al_2_O_3_, SiO_2_, TiO_2_, etc. Alternatively, char (activated char/carbon) obtained from biomass could be employed as a low-cost catalyst support and environmentally benign material to replace the metal oxides, which are expensive and have other inexhaustible industrial applications. Li et al. [9] investigated the catalytic performance of Fe, Ni and Co catalysts supported on γ-Al_2_O_3_ and activated carbon during the pyrolysis-catalysis conversion of waste tires into CNTs and H_2_. Their study revealed that activated carbon had better catalytic performance than γ-Al_2_O_3_; all activated carbon-supported catalysts presented the highest catalytic activity with better quality CNTs and high H_2_ yield compared to γ-Al_2_O_3_ supported catalysts. Gong et al. [17] studied the production of CNTs using a one-pot carbonization of PP over a combined catalyst of activated carbon and Ni_2_O_3_ (AC/Ni_2_O_3_); they reported that the AC content played a significant role due to the good synergistic catalytic effect in the yield of CNTs. In their study, they showed that the carboxylic groups (surface functional groups) present in the AC promoted the cracking of PP into light hydrocarbons and further dehydrogenation, and there was also an in situ reduction of Ni_2_O_3_ to metallic Ni as well as aromatization of light hydrocarbons intermediates to form CNTs.

Corn is one the most widely cultivated crops in most parts of the world. Corn cobs are lignocellulose biomass obtained from corn plants after their grains are removed; they amount to about 20% of the corn residue [18]. In South Africa, corn cobs are one of the largest agricultural wastes; there are over 9 million tonnes of corn cob wastes yearly [19,20]. These wastes after been harvested are incinerated, which results in increasing carbon footprints and pollution (land and air), and this also triggers global warming and other environmental and economic concerns. However, several studies have been devoted to various possible uses of corn cob as adsorbents, bioethanol, chemicals, etc., but limited work so far has researched char (activated carbon) as catalyst supports for CNTs production. Guizani, Sanz and Salvador [21] evaluated the catalytic activities of raw biomass char and Ni-supported biomass char during methane decomposition for H_2_ production and reported that Ni-supported biomass char exhibited 10 times better catalytic activity than raw char during methane cracking to H_2_.

CNTs growth especially from carbonaceous materials are usually hindered by catalyst deactivation, which reduces the catalyst life during CNTs synthesis; hence, the presence of promoters could enhance the catalyst life and catalytic activity. Corn cob-derived char has been shown to contain a good percentage of potassium (alkali metal), which could act as a promoter in catalysis [22]. Catalyst deactivation arising from coke formation/deposition on the surface of a catalyst could be reduced by promoters. Furthermore, the presence of alkali-earth metal and other metals such as magnesium, phosphorus, silicon, etc. usually present in biomass-derived char could result in the formation of spinel, which is also a good catalyst support precursor [22]. 

Different modification/activation techniques such as chemical (alkaline and acid), physical and combined modification/activation methods have been widely studied to improve the surface properties of biochar [23]. In alkali activation, KOH, K_2_CO_3_, NaOH, etc. are the most widely used reducing agents, while the physical activation method involves the use of air, steam or CO_2_ to facilitate the devolatilization and formation of pure carbon in the biochar [24,25]. Song et al. [26] compared the characteristics of chemically and physically activated carbon obtained from corn cob. They reported that chemically activated carbon obtained from corn cob demonstrated a higher specific surface area and larger pore volume than the physically (steam) activated carbon. Nonetheless, they concluded that the amount of activating agent also played a pivotal role in the formation of pores and affected the surface characteristics of biochar. 

Under high temperatures around 800 °C, the KOH (or carbonates) activating agent decomposes completely to produce metallic potassium, which supports and strengthens pore expansion in the carbon matrix, resulting in the release of CO_2_, CO and H_2_ [26,27,28], as shown in the reactions below:6KOH + 2C → 2K_2_CO_3_ + 2K + 3H_2_(1)

At high temperature above 500 °C, metal alkali carbonate decomposes, and the reaction continues as follows:K_2_CO_3_ + C → K_2_O + 2CO(2)
K_2_CO_3_ → K_2_O + CO_2_(3)
2K + CO_2_ → K_2_O + CO(4)

At about 800 °C and above, the reaction ensues as:K_2_O + C → 2K + CO(5)

In this study, we utilized waste corn cob-derived biochar as a catalyst support for the bimetallic NiMo catalyst in the synthesis of carbon nanotubes using waste polypropylene plastics as carbonaceous feedstock in a single-stage chemical vapor deposition reactor. The focus is on investigating the effect of different biochar activation methods on the microstructure, quality, and the yield of produced CNTs obtained from waste PP. Char activation in catalysis has been recognized as a fundamental step toward improving the surface properties and the development of desired porous structure to enhance catalytic activity. Several studies in the literature have utilized NiMo-based catalysts and agro-based derived carbon as catalysts [29] and support catalysts [30] in nanomaterials synthesis. However, no study has overtly addressed the influence of various char activation techniques on the yield, quality and morphology of CNTs produced or established the link between activation and the catalytic performance in nanomaterials synthesis especially when using polymer feedstock such as waste plastics in CNTs synthesis.

## 2. Materials and Methods

### 2.1. Carbonization of Biomass

A corn plant was received from a small farm in Rooderport, Gauteng, South Africa. The obtained corn plant was deseeded, washed with deionized water, and dried under the sun for 48 h to lessen the moisture content. The dried corn cob was pulverized and sieved using 50–100 mesh particle size. The obtained granular sized corn cob was pyrolyzed in a horizontal tube furnace at 800 °C at a 20 °C/min heating rate for 240 min under argon (99% purity) atmosphere. The obtained char was treated with 0.1 M hydrochloric acid (32%) and further washed with warm and cool deionized water until the pH of the filtrate solution was 7.

### 2.2. Char Activation 

For the activating agent, 85% purity potassium hydroxide (Rochelle, Johannesburg, South Africa) was used according to the procedure described by [31]. Some amount of the char was impregnated with a saturated solution of KOH (4 M) at a char to chemical ratio of 1:3 based on mass. Obtained samples were oven-dried overnight at 110 °C. The dried sample was afterwards heated to 800 °C for 120 min in a horizontal tube furnace under argon (99% purity) atmosphere at a heating rate of 20 °C/min. The obtained KOH-activated biochar was labeled AC_X_. 

A similar step was followed for steam physical activation; generated steam from the heating mantle was introduced into the horizontal tube furnace housing the char under similar conditions (800 °C at 20 °C for 120 min under Ar) and designated as AC_T_. 

In addition, biomass was directly carbonized, treated with 0.1 M HCl and further washed with deionized water until the pH was 7 and dried in the oven overnight at 110 °C. Afterwards, it was subjected to thermal treatment at 800 °C for 120 min in a horizontal tube furnace in an Ar purged environment without any additional KOH chemical or steam physical activation and hence labeled as AC_0_. 

### 2.3. Preparation of Catalysts

Different NiMo catalysts supported on KOH-treated char (AC_X_), steam-activated char (AC_T_), and non-activated char (AC_0_) were prepared by impregnation method. We calculated the amount of Ni and Mo precursors; Ni(NO_3_)_2_·6H_2_O (Rochelle) and (NH_4_)_6_ Mo_7_O_24_·4H_2_O salts (Rochelle, South Africa), respectively, were used to achieve 10 wt % Ni loading and a molar ratio of Ni to Mo at 5:1. These precursor salts were firstly dispersed in deionized water, and the resultant mixture was impregnated on 1.5 g of each active carbon support (AC_X_, AC_T_ and AC_0_) under vigorous stirring for 6 h. The obtained mixture was dried in an oven for 12 h at 110 °C. The resultant solid was ground (using an agate mortar and pestle) and calcined at 600 °C for 3 h under air. The three catalysts obtained were designated as NiMo/AC_X_, NiMo/AC_T_ and NiMo/AC_0_ based on the activation method used.

### 2.4. Catalyst Testing (Synthesis of CNTs)

Waste PP materials (household food packaging wastes) were obtained from the refuse deposition and collection site of the University of Johannesburg, Doornfontein Campus, Johannesburg, South Africa. Collected wastes were washed, sun-dried and cut into small pieces using a pair of scissors. The experimental set-up and procedure for synthesizing CNTs from waste PP using a single-stage chemical deposition technique have been described in detail elsewhere [32]. Briefly, the CVD set-up consists of a quartz tube reactor of about 110 cm length and 50 mm I.D fit into a horizontal tubular heated furnace. About 1.0 g of calcined catalyst was loaded on a ceramic boat positioned at the middle of the reactor and the set-up temperature was regulated to 700 °C at a heating rate of 10 °C/min. Once the fixed temperature was attained, in situ catalyst reduction was initiated for 40 min under H_2_/Ar gas mixture (5 vol % H_2_/95 vol % Ar) at the flow of 120 mL/min. Afterwards, the H_2_/Ar gas mixture was switched to Ar gas (99% purity) while still maintaining the gas flow at 120 mL/min. Subsequently, about 1.0 g of cut waste PP loaded in another ceramic boat was pushed in and kept at the region where the temperature was around 450 °C, that is, some distance away from the location of the catalyst. The set-up was allowed until complete PP feedstock decomposition, and the subsequent deposition and growth of carbon nanomaterial on the catalyst was for about 30 min. At the end of the synthesis reaction, the apparatus and its contents were cooled down overnight under an inert environment. The experiment was repeated three times under similar conditions to demonstrate the reliability and repeatability of the results, and the presented data were average values of the repeated experimental results. The obtained material was purified using 0.1 M HCl mild acid treatment and later washed with deionized water until the pH is neutral. The yield of carbon produced was obtained with respect to the activity of the catalyst as described by refs. [33,34].

### 2.5. Characterization of Catalyst and CNTs

The prepared catalysts were characterized using a variety of characterization methods. 

The metal phase structures present were determined by X-ray diffraction (XRD) by means of a Rigaku Ultima IV X-ray diffractometer (Rigaku Co., Tokyo, Japan) with Cu Kα radiation in the 2θ range from 10° to 90°. 

The BET surface area of the catalysts was determined by the nitrogen adsorption experiments using ASAP 2020 Surface Area Analyzer (Micromeritics, Atlanta, GA, USA).

The morphology and elemental composition of char, corncob and catalysts were obtained by scanning electron microscope coupled with energy-dispersive X-ray spectrometer (SEM-EDXS) using VEGA 3 TESCAN (TESCAN, Brno, Czech Republic). 

The microstructures of deposited carbon on the surfaces of catalysts were determined by JEM-2100 transmission electron microscope (JEOL, Tokyo, Japan). 

Functional groups on the surfaces of prepared catalysts and synthesized carbon were detected by Fourier Transform Infrared (FTIR, spectrum 100 FTIR spectrometer, PerkinElmer, Waltham, MA, USA) using the KBr pellet technique. The spectra were recorded in the range of 500–4500 cm^−1^.

The graphitic quality of the deposited carbon was determined by Raman spectroscopy using a WiTec focus innovations Raman spectrometer (WiTec, Ulm, Germany) with Raman shift from 1000 to 3000 cm^−1^ at 532 nm excitation wavelength equipped with a diode Nd: YAG laser at room temperature. Samples were exposed for 20 s under a power laser output of 20 mW.

### 2.6. Ultimate and Proximate Analysis of Corn Cob, Char, and Waste PP 

The ultimate (carbon, hydrogen, nitrogen, and sulfur) analyses of corn cob (biomass), char and waste PP were carried out using a CHNS1000 Elemental analyzer (Leco, St. Joseph, MI, USA). Proximate (moisture, volatile matter, fixed carbon, and ash content) analyses were performed according to the American Society for Testing and Materials (ASTM) standard as reported by Budai et al. [35].

## 3. Results and Discussion

### 3.1. Characterization of Biomass (Corn Cob) and Biochar 

The proximate and ultimate analyses results for biomass, biochar and waste PP are shown in Table 1. The proximate analysis showed that the fixed carbon content of char increased compared to the biomass, hence its suitability as a catalyst support. Waste PP showed considerably high volatile matter content depicting the presence of a greater composition of low molecular weight hydrocarbon, which is advantageous in utilizing waste PP as a cheap carbon feedstock for growing carbon nanomaterials. Furthermore, the high-volatile matter content of the biomass (corn cob) could also be linked to the easy bond breaking of the cellulose, hemicellulose and lignin components in the corn cob biomass, resulting in the release of low molecular weight hydrocarbon gases [36,37]. The ash content depicts the inorganic mineral content in biomass, biochar and waste PP.

The thermochemical decomposition of corn cob biomass resulted in increased inorganic composition with biochar content higher than corn cob inorganic content, as shown in Table 2. Table 2 shows the elemental compositions of biomass and biochar. From the result, the obtained carbon and potassium contents increased after thermochemical processing of the biomass into biochar, and the appearance of other metals is also observed. The compositional characteristics of corn cob and the corresponding char are reported to be affected by the soil type, type of soil nutrient applied, region where the corn plant was cultivated, etc., which could result in a variation in results obtained for biochar in relation to other reported findings from other researchers [36,38]. According to Srilek and Aggarangsi [39], high-temperature pyrolyzed corn cob biomass (biochar) has high lignin content and completely decomposed cellulose and hemicellulose contents, which could influence the overall inorganic composition and surface properties. 

Scanning electron microscopy was conducted to study the morphology of corn cob and biochar (Figure 1). Biochar (Figure 1B) showed a porous, rough surface structure with almost evenly distributed vascularized cavities. Hu et al. [23] also reported a similar structure for corn cob-derived biochar. The biomass surface structure (Figure 1A) showed more of a compacted mass structure with few pores.

### 3.2. Characterization of Catalysts and Biochar

Scanning electron microscopy was conducted to study the morphology of all prepared carbon-supported NiMo catalysts (Figure 2). The non-activated biochar-supported NiMo catalysts depicted in Figure 2a showed a catalyst structure with almost similar particle sizes, while 2b and 2c displayed a spongy-like structure. EDXS showed the presence of all elemental components of the NiMo/AC catalyst under study (see Supporting Information: Appendix A)

A nitrogen adsorption–desorption test was carried out at 77 K to determine the surface properties of all catalysts, as shown in Figure 3. According to the International Union of Pure and Applied Chemistry (IUPAC) classification [40], the isotherms of all catalysts under study fall under type IV class with an H3 hysteresis loop observed at a relative pressure (P/P_0_) around 0.72–0.76, suggesting the presence of large mesopores. The isotherm, type IV is usually associated with materials whose geometry is mesoporous. However, materials are normally considered mesoporous when there is no overlapping adsorption–desorption hysteresis at high relative pressure. Based on the three major pore size classifications of porous materials—micropores (<2 nm), mesopores (2–50 nm) and macropores (>50 nm)—all prepared catalysts showed mesoporous characteristics with pores sizes between 2 and 50 nm, as shown in Table 3. Examining the effect of activation on the surface properties of char, a significant increase in BET surface area was observed for all activated char, as shown in Table 3. However, all catalysts showed a reduction in the BET surface area upon impregnation with NiMo. The catalyst supported on steam-activated char gave the least surface area compared to chemical-activated char; this could be due to pore blockage originating from the metal catalysts [41]. The obtained specific surface areas of char and modified char are different. However, activation improved the specific surface area of carbonized char, as shown in Table 3. As pointed out in the literature, the surface characteristics of alkaline-modified biochar depend on different factors such as the amount of activating agent, the source of feedstock, soil type and texture, type of nutrient added to the plant, preparation method, overall cellulose, lignin and hemicellulose composition, etc. [24,42]. Therefore, different activating methods have a substantial effect on the surface properties of the catalysts. 

FTIR analysis was undertaken to identify the functional groups present on the catalysts’ surface. The IR spectra of all NiMo catalysts prepared using corn cob char support obtained from different activation techniques are shown in Figure 4. Observed spectra display various broad, weak, strong and medium peaks intensities resulting from different band activities. Major functional groups were observed between 500 and 2000 cm^−1^ due to the presence of different functional groups arising from the C-O stretching of alcohols, esters, ethers and phenol [43]. This is similar to other reported catalysts prepared from lignocellulosic corn cob precursors [23,44,45]. The intense peak at 956 cm^−1^ is attributed to the aromatic C-H bending; the intensity of this band is higher in NiMo/AC_T_ compared to NiMo/AC_0_ and NiMo/AC_X_. The broad peak at 3430 cm^−1^ is due to the stretching vibrations arising from the hydroxyl (-OH) functional groups from phenol, carboxyl or alcohols [37,43,44]. These peaks become weaker in the NiMo/AC_0_ catalyst, which is suggestive that activation resulted in a structural change with the emergence of functional groups on the catalysts’ surface. 

Figure 5 depicts XRD patterns of NiMo/AC catalysts activated using different methods. All phases of interest were obtained; also, several phase overlaps were observed in all the prepared catalysts. No K-containing phases were identified in the chemically activated catalyst, indicating that K-containing phases could be either too small to be detected or they may have been well dispersed on the catalyst surface [46]. It could be observed that there are no NiO phases in NiMo/AC_T_ (which could also be because they are too small to be detected); rather, two forms of NiMoO_4_ crystal phases with similar crystal structures were majorly present in the NiMo/AC_T_ catalyst: a monoclinic structure of metastable β-NiMoO_4_ (ICDD 00-045-0142) and monoclinic structure of α-NiMoO_4_ (ICDD 01-086-0361). However, α-NiMoO_4_ phases were only detected in NiMo/AC_0_ and NiMo/AC_X_ catalysts; this result showed that steam activation resulted in the emergence of more metastable crystal phases on the surface. Other detected phases were MoOC (ICDD 00-017-0104), CO (ICDD 01-074-1229) and hexagonal NiO (ICDD 00-022-1189). 

### 3.3. Effect of Activation Methods on the Microstructure and Morphology of Synthesized Carbon Nanomaterial 

The TEM images in Figure 6 confirmed that all catalysts under study were active in the formation of different kinds of carbon nanomaterials (CNMs) from waste PP in a single-stage CVD technique. CNT_0_ deposited on the surface of the NiMo/AC_0_ catalyst showed several bamboo knots-like hollow partitioned structures known as bamboo-like types of CNTs [47] with a mean outer diameter within 12–36 nm and length up to several microns. Carbon deposits (CNT_X_) on the surface of the NiMo/AC_X_ catalyst also showed bundles of entangled filamentous carbon structures confirmed to be multiwalled carbon nanotubes (MWCNTS), with the mean outer diameter in the range of 10–40 nm. Interestingly, the as-synthesized nanomaterial (CNT_T_) deposited on the surface of NiMo/AC_T_ displayed a well-graphitized hollow filamentous multiwalled structure confirmed to be MWCNTs with a mean tube outer diameter between 10 and 30 nm. Similarly, several amorphous carbons and lots of metal residues could be observed (see Appendix A). It is observed that both the base-growth and tip-growth models were prevalent in synthesized nanotubes, depicting the interaction between the catalyst and support. The base-growth model where the active metal particles are trapped along the hollow core of the nanotubes is observed more in CNT_0_. The obtained interlayer spacings between graphitic layers of CNT_0_, CNT_X_ and CNT_T_ were 0.335, 0.335 and 0.34 nm respectively, which is consistent with the model graphitic interlayer spacing suggesting a higher graphitization degree of all synthesized CNTs. Oliveira, Franceschini and Passos [48] relate catalyst selectivity toward a particular kind of nanomaterial to the active metal particle size, which in turn is dependent on the support textural properties.

### 3.4. Influence of Activation Methods on the Quality and Yield of Deposited CNTs

In other to evaluate the quality/purity and graphitization degree of synthesized CNTs, Raman spectroscopy was undertaken, and the Raman spectra are shown in Figure 7. Three distinctive peaks are seen at 1350, 1580 and 2678 cm^−1^ wavelengths for all the carbon nanomaterials produced. The peaks at 1580 cm^−1^ corresponding to the G-band are associated with an E_2g_ mode of hexagonal graphitic carbon and are related to the vibration of sp^2^-bonded carbon atoms within the graphite sheet [49]. The peak at 1350 cm^−1^ corresponds with the D-band and is associated with defects or disordered graphite within the graphene lattice; the 2D or G’ band at about 2678 cm^−1^ describes the two-photon elastic scattering process, which is attributed to the overtone of the D-band. The intensity ratio of G/D (I_G_/I_D_) is employed to estimate the graphitization degree of the deposited carbon nanomaterial. A larger I_G_/I_D_ value signifies a higher structural ordering for CNTs, which also describes the higher quality and purity of CNTs [50]. The Raman spectra in Table 4 indicated that the I_G_/I_D_ ratio of CNT_T_ is higher than CNT_0_ and CNT_X_ with values of 1.44, 1.09, and 1.08, respectively, suggesting that CNT_T_ has more graphitic and structurally ordered CNTs with fewer defects than CNT_0_ and CNT_X_. This result is in good agreement with the TEM result. The majority of metal-stable phases of α- and β-NiMoO_4_ phases [51] identified in the XRD result could be the reason for the better graphitization degree obtained in CNT_T_, since they assist in the stabilization of the active phases on the supports by averting excessive agglomeration [52]. To further confirm the ordering of all synthesized CNTs, deconvolution of the Raman spectra was undertaken using the Lorentzian function. There is no evidence of the D’ band, which is typical of the disordered/defective graphitic structured CNTs in all of the as-synthesized materials, as shown in the deconvoluted Raman spectra in Appendix A. A recent study by Shah et al. [53] reported an I_G_/I_D_ value of 0.83 to 0.94 for CNTs synthesized using biochar as a catalyst and polypropylene copolymer as a feedstock at catalytic CVD temperatures of 500 to 900 °C. Hence, higher quality and well-graphitized CNTs were obtained from this present study. It could be speculated that different char activation methods significantly influenced the purity and quality of CNMs obtained from waste PP. 

The yield of obtained CNTs was estimated based on the mass of carbon deposited per mass of catalyst used, as depicted in Table 4. CNT_X_ deposited on the surface of NiMo/AC_X_ produced the highest yield amongst synthesized CNTs, while CNT_T_ gave a very low yield. This result is consistent with reported findings from the literature on the trade-off between the yield and purity of CNTs [54]. Therefore, chemical activation resulted in an improved CNTs yield, while the physical (steam) activated catalyst gave a very low CNTs yield. The presence of metallic K (see Appendix A of the Supporting Information) in the catalyst could act as porogens (pore-forming agents) and therefore at high-temperature treatment conditions (catalyst reduction and synthesis) could result in carbon methanation providing an abundance of lighter hydrocarbons fractions (such as CH_4_, CO, etc.) needed for nanomaterial growth [37] and, hence, improved nanomaterial yield. One can infer that chemical activation supported and strengthened pore expansion in the carbon matrix, resulting in the formation of intermediate compounds such as CO and CO_2_ promoting an additional source of carbon, as observed in Equations (1)–(5) and possible carbon methanation as a source of methane (CH_4_). Furthermore, the formation of excessive metal-stable species [52] as seen in the NiMo/AC_T_ could result in the reduction of accessible active metal phases, which are accountable for the nucleation and growth of CNTs, resulting in a low CNTs yield (CNT_T_). CNT_0_ deposited on the surface of a non-activated char catalyst also gave a high CNTs yield. Nevertheless, either activation technique may not be necessary for corn cob-derived char in the synthesis of high-yield CNTs from waste PP. 

FTIR was used to study the different surface functional groups formed in CNT_0_, CNT_X_ and CNT_T_. Figure 8 shows the FTIR spectra; the broad peak at 3434 and 3435 cm^−1^ originating from the stretching vibrations of hydroxyl (-OH) group is indicative of absorbed water molecule on the sidewalls of CNMs. Asymmetric/symmetric methyl stretching bands at 2913 cm^−1^ [55,56] are also observed for CNT_0_ and CNT_X_; typically, these groups are assumed to be located at defect sites on the sidewall surface, resulting from the cleavage of ether functional group during refluxing in acid (during CNMs purification and water molecules during washing) [57]. The peaks at 1629, 1636 and 1639 cm^−1^ could be attributed to the C=C stretching vibration, which indicates the skeletal graphite structure of CNTs. The peaks at 1392 and 1393 cm^−1^ can be assigned to the C–O stretching typical of the carbonyl of quinone type unit at the sidewalls of the CNTs. Peaks at 1084 and 1116 cm^−1^ [56] can be assigned to the C–H vibration mode of aromatic, carboxylic, ether, phenol or ester groups arising due to the mild acid refluxing during the purification of synthesized nanotubes. However, in this region, due to band overlapping at the defect sidewalls, it is difficult to assign these peaks to a particular vibrational mode. The peak region at 500–1000 cm^−1^ reveals the existence of asymmetrical hexagonal carbon at the sidewalls due to an attack on the carbon double bond during acid treatment and purification [58]. 

## 4. Discussion

This study has shown the influence of different activation techniques on the biochar used as a low-cost catalyst support for the bimetallic NiMo catalyst as well as their effect on the quality, yield and morphology of deposited CNTs obtained from waste polypropylene plastics. The studied biochar was obtained from the pyrolysis of waste corn cob and was activated by chemical and physical methods using KOH and steam, respectively. Analysis of the results obtained from this study essentially revealed the correlation between the different char treatment (activation) methods on the surface properties of prepared catalysts and their corresponding catalytic activity/performance in CNTs synthesis from waste PP plastics. Significant differences regarding the yield and quality of deposited CNTs on the surfaces of KOH and steam-activated catalysts were observed. 

The emergence of numerous meta-stable phases on the surface of steam-activated char supported catalyst (NiMo/AC_T_) impacted the yield with a very low amount of CNTs (about 70 mg CNTs/g catalyst) compared to the amount of CNTs deposited over the surface of chemically activated char-supported catalyst (NiMo/AC_X_). The excessive metal-stable species in the NiMo/AC_T_ could lessen the available active metal phases [52], which are bedrocks for the nucleation and growth of CNTs, hence resulting in low CNTs yield (CNT_T_). These phases also resulted in the stabilization of metal active phases while preventing excessive aggregation/agglomeration. Consequently, a higher graphitization degree of CNTs obtained over the NiMo/AC_T_ catalyst was observed with an intensity ratio of G/D: I_G_/I_D_ = 1.44. The improved yield of CNTs obtained over KOH-activated char supported catalyst (NiMo/AC_X_) could be due to the presence of metallic K in the catalyst. This acts as pore-forming agents, which support and strengthen pore expansion in the carbon matrix, resulting in the formation of intermediate compounds such as CO and CO_2_ and promoting an additional source of carbon. Therefore, at high temperature treatment conditions (during catalyst reduction and synthesis), carbon methanation occurs, resulting in an abundance of lighter hydrocarbons fractions (such as CH_4_, CO, etc.) [37], which are needed for nanomaterial growth and hence improved nanomaterial yield. KOH-activated char-supported catalyst resulted in a high CNTs yield of about 470 mg CNTs/g catalyst. Earlier studies by Zhang et al. [29] observed that the K element detected in hollow carbon nanofibers (HCNFs) played a catalytic role during the formation and growth of HCNFs. They concluded that the K may have originated from either the pine nut shell (PNS) biomass or from activated carbon (AC) obtained from microwave pyrolyzed char. However, this study did not clearly highlight the role played by the presence of K in the formation and nucleation of the nanomaterial. Therefore, this present study has been able to provide new knowledge on the role played by K as the activating agent in improving the yield of produced CNTs from waste PP plastics. Furthermore, activation resulted in the formation of filamentous CNTs with varying diameters and lengths, while the non-activated char supported catalyst (NiMo/AC_0_) resulted in bamboo-like CNTs (BCNTs).

Table 5 summaries the result obtained from this study with some reported studies in the literature utilizing a similar active NiMo catalyst but different catalyst supports, with similar or different polymer as carbon feedstock. Higher graphitized and quality CNTs were produced in this study compared to all reported CNTs quality in the literature as shown. In addition, the defectiveness of CNTs obtained in this study was better compared to what was reported in some of the studies. 

Further research studies based on the results of this study may be undertaken to explore the influence of activating conditions, the amount of activating agent, and possibly, employ other activating agents such as air, CO_2_, acids and other alkaline precursors (K_2_CO_3_, NaOH, etc.) in optimizing the viability of waste corn cob as a low-cost catalyst in producing high value-added CNTs with tailored characteristics from waste plastic polymers.

## 5. Conclusions

Carbon nanotubes were successfully generated by the catalytic decomposition of waste PP in a single-stage reactor using biomass waste-derived catalytic material obtained from corn cob as the catalyst support. Activating agents (in this case, KOH, and steam) were highly related to the surface properties and crystallinity of the catalysts. Under different support activating agents, different morphologies of CNTs were obtained; for example, bamboo-like structured multiwalled carbon nanotubes were obtained using the NiMo catalyst supported on non-activated char (NiMo/AC_0_), while NiMo catalysts supported on KOH (chemical)-activated char (NiMo/AC_X_) and steam-activated char (NiMo/AC_T_) resulted in filamentous long, rugged surface MWCNTs. Again, high-quality CNTs were deposited over the NiMo/AC_T_ catalyst with very low yield estimated to be about 70.0 mg CNTs/g catalyst compared to CNTs deposited over NiMo/AC_0_ and NiMo/AC_X_ with a yield of 430.0 and 470.0 mg CNTs/g catalyst, respectively. The low CNTs yield obtained with the NiMo/AC_T_ catalyst could be correlated to the dominance of meta-stable (α- and β-NiMoO_4_) phases in the catalysts, as observed in the XRD result, signifying that activation affects the surface properties of the catalysts, which influenced the microstructure, type, and yield as well as quality of CNTs obtained. Therefore, it could be concluded that chemical activation promoted increased CNTs yield while steam (physical) activation supported better quality (purity) CNTs, depicting a trade-off between CNTs quality and yield over steam and chemical activation methods. Nevertheless, the physical char activation technique with steam may not always be adopted where high-yield CNTs are required. 

## Figures and Tables

**Figure 1 polymers-14-02898-f001:**
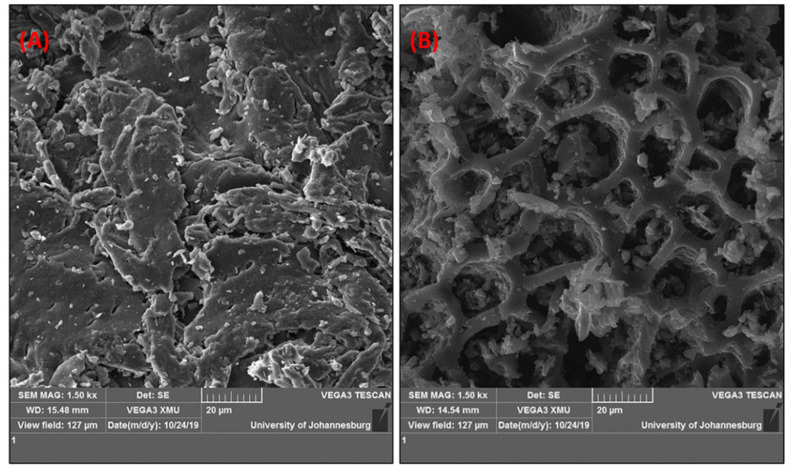
SEM micrographs of corn cob (**A**) and biochar (**B**).

**Figure 2 polymers-14-02898-f002:**
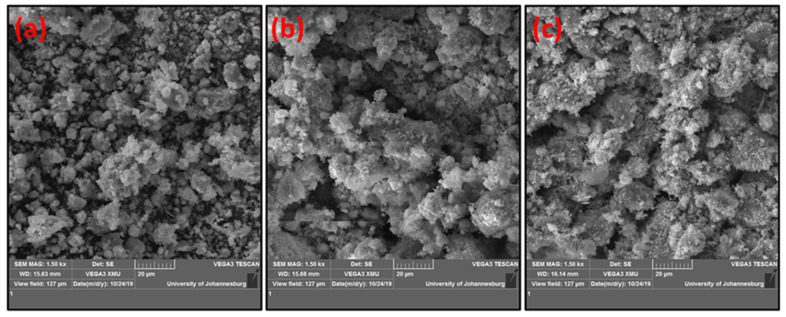
SEM images of prepared catalysts using different activation methods: (**a**) inactivated NiMo/AC_0_, (**b**) NiMo/AC_X_, (**c**) NiMo/AC_T_.

**Figure 3 polymers-14-02898-f003:**
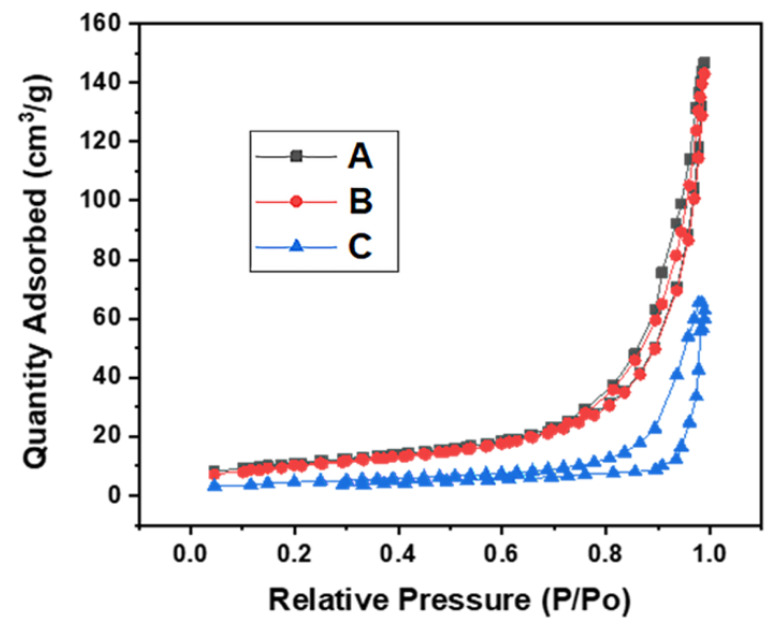
N_2_ adsorption–desorption isotherms of prepared NiMo/AC catalysts: (**A**) NiMo/AC_X_, (**B**) NiMo/AC_0_ and (**C**) NiMo/AC_T_.

**Figure 4 polymers-14-02898-f004:**
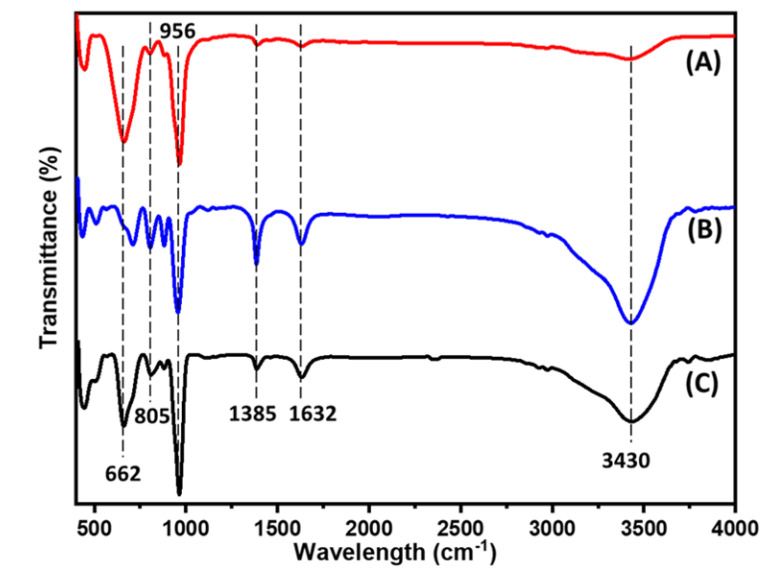
FT-IR spectra of all catalysts prepared using char supports activated by different char act-vation methods: (**A**) NiMo/AC_0_, (**B**) NiMo/AC_X_ and (**C**) NiMo/AC_T_.

**Figure 5 polymers-14-02898-f005:**
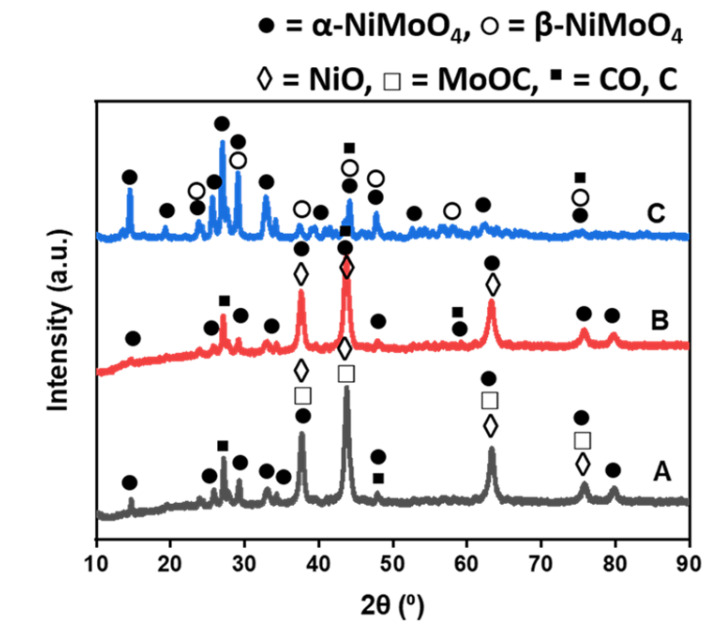
XRD pattern of prepared catalysts using different char activation methods: (**A**) NiMo/AC_X_, (**B**) NiMo/AC_0_ and (**C**) NiMo/AC_T_.

**Figure 6 polymers-14-02898-f006:**
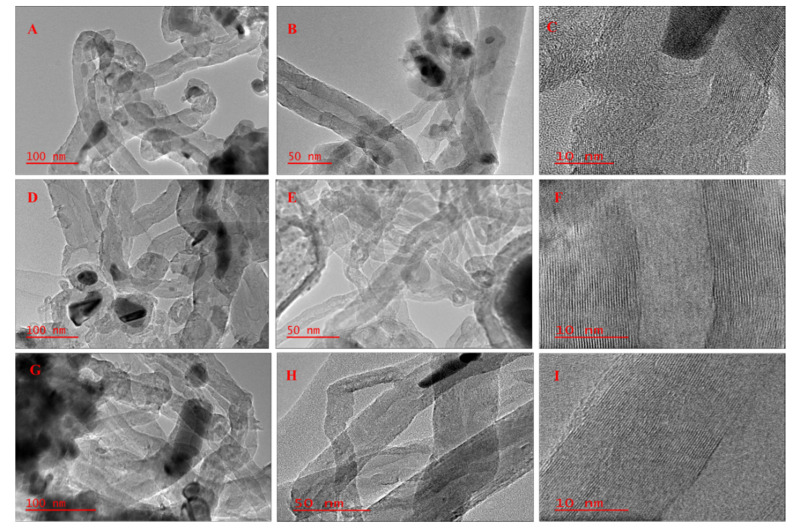
TEM micrographs of synthesized CNMs. (**A**–**C**) CNT_0_; (**D**–**F**) CNT_X_ and (**G**–**I**) CNT_T_.

**Figure 7 polymers-14-02898-f007:**
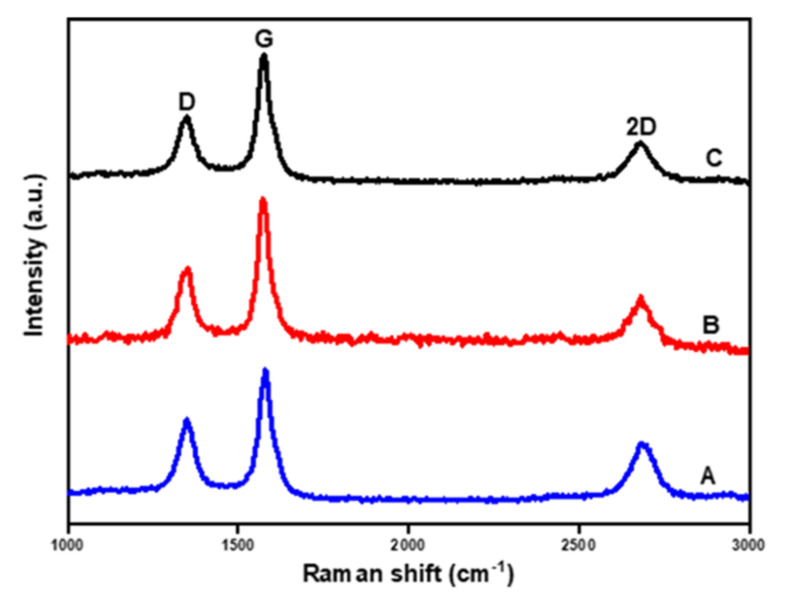
Raman spectra of as-produced CNMs: CNT_X_ (**A**), CNT_0_ (**B**) and CNT_T_ (**C**).

**Figure 8 polymers-14-02898-f008:**
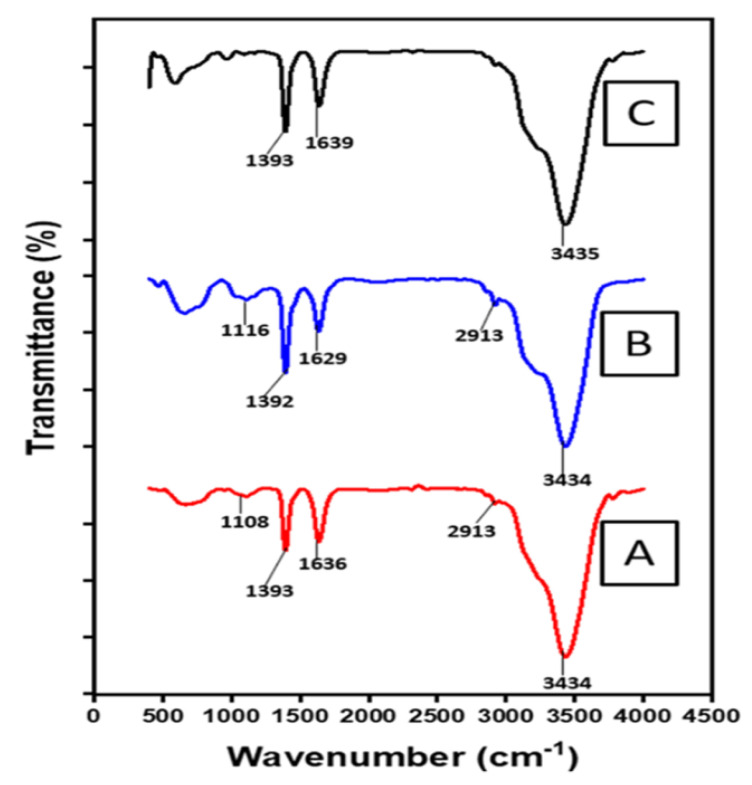
FT-IR spectra of deposited CNTs: (**A**) CNT_0_, (**B**) CNT_X_ and (**C**) CNT_T_.

**Table 1 polymers-14-02898-t001:** Proximate and ultimate analysis of corn cob, biochar, and waste PP.

	Proximate Analysis				Ultimate Analysis				
	Moisture (%)	Ash * (%)	Fixed Carbon (%)	Volatile Matter (%)	Nitrogen (wt %)	Carbon (wt %)	Hydrogen (wt %)	Sulfur (wt %)	Oxygen * (wt %)
Corn cob	5.9	1.0	14.0	79.1	0.14	44.85	5.87	ND	49.14
Biochar	5.2	4.5	79.6	10.7	0.35	87.87	0.97	0.57	10.24
Waste PP	1.5	0.4	0.0	98.1	0.11	83.43	13.67	ND	2.78

* By difference; ND—Not detected; Elemental analysis are given in ash-free basis.

**Table 2 polymers-14-02898-t002:** Elemental compositions of biomass and biochar.

Composition	Corn Cob (wt %)	Biochar (wt %)
C	57.68	88.43
O	41.07	8.41
Si	0.12	0.39
Cl	0.15	0.12
K	0.79	2.15
Ni	0.18	0.16
Mg	-	0.08
P	-	0.14
S	-	0.11

**Table 3 polymers-14-02898-t003:** Surface characteristics of corn cob-activated chars and NiMo/AC catalysts.

Catalyst	BET Surface Area (m^2^ g^−1^)	Pore Volume (cm^3^ g^−1^)	Pore Size (nm)	NiO * Crystallite Size (nm)
Char	0.78 ± 0.06	0.004	19.0	-
AC_0_	38.13 ± 0.02	0.101	28.11	
AC_X_	48.15 ± 0.22	0.350	23.20	-
AC_T_	41.70 ± 0.14	0.281	24.19	-
NiMo/AC_X_	35.87 ± 0.06	0.227	25.82	12.3
NiMo/AC_T_	16.74 ± 0.16	0.349	19.72	16.0 ^@^
NiMo/AC_0_	21.02 ± 0.14	0.255	24.83	8.3

* NiO crystallite size was obtained from XRD; ^@^ NiMoO_4_ (α- and β-) crystallite size.

**Table 4 polymers-14-02898-t004:** Yield and quality of all synthesized CNMs.

Notation	Quality I_G_/I_D_	Defect I_D_/I_G_	Yield (mg CNTs/g Catalyst)
CNT_0_	1.09	0.92	430.0
CNT_X_	1.08	0.92	470.0
CNT_T_	1.44	0.69	70.0

**Table 5 polymers-14-02898-t005:** Comparison of results obtained from this study with those reported in the literature.

Catalyst	Feedstock	Synthesis Method/Condition	Peak Intensity Ratio	Yield	Refs.
Ni/Mo/MgO	PP	Autoclave, 800 °C	I_G_/I_D_ = 0.75–0.93	3.2 g CNTs/6 g PP	[59]
Ni/Mo/MgO	HDPE	Multi-core reactor, 700–800 °C	I_G_/I_D_ = 0.69–0.99	6.03 g CNTTs/30 g HDPE	[60]
NiMo/Al_2_O_3_	LDPE	Two-stage CVD, 500–700 °C	I_D_/I_G_ = 0.93–2.11	14.7–28.1% (% weight of polymer used)	[61]
NiMo/MgO	PP	One-stage CVD, 700 °C	I_G_/I_D_ = 0.99	33.3%	[62]
NiMo/CaO	PP	One-stage CVD, 700 °C	I_G_/I_D_ = 1.00	31.0%	[62]
NiMo/TiO_2_	PP	One-stage CVD, 700 °C	I_G_/I_D_ = 1.07	37.0%	[62]
NiMo/CaTiO_3_	PP	One-stage CVD, 700 °C	I_G_/I_D_ = 1.25	40.0%	[62]
NiMo	PP	One-stage CVD, 700 °C	I_G_/I_D_ = 0.93	18.4%	[62]
NiMo/ AC_0_	PP	One-stage CVD, 700 °C	I_G_/I_D_ = 1.09; I_D_/I_G_ = 0.92	430 mg CNTs/g catalyst	Present Study
NiMo/ AC_X_	PP	One-stage CVD, 700 °C	I_G_/I_D_ = 1.08; I_D_/I_G_ = 0.92	470 mg CNTs/g catalyst	Present Study
NiMo/ AC_T_	PP	One-stage CVD, 700 °C	I_G_/I_D_ = 1.44; I_D_/I_G_ = 0.69	70 mg CNTs/g catalyst	Present Study

## Data Availability

Not applicable.

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
