# Peer review of "Corn Cob Char as Catalyst Support for Developing Carbon Nanotubes from Waste Polypropylene Plastics: Comparison of Activation Techniques"

_polymers, 2022, doi:10.3390/polym14142898_

Round 1

Reviewer 1 Report

Polymers-1678421

This manuscript presents an exciting approach to using renewable feedstock for developing carbon nano-2 tubes from waste polypropylene plastics. The authors describe different activation procedures that would influence the final nanotube structure. They performed different characterization techniques to show the effect of activation protocols on the properties of the catalyst precursor.

              However, the present form requires some additional work prior be accepted for publication

This reviewer encourages the authors to take into consideration the following

  1. The quality of readiness of the manuscript must be improved.
  2. There are some examples of better uses of English in the PDF manuscript
  3. The introduction can be improved by presenting the critical relevant information to introduce the objectives of this research to the readers.
  4. The authors should document how they know that the Ni/Mo ratio or concentration was achieved.
  5. There are other comments across the document. Please address comments or questions accordingly

Author Response

Open review 1

Comment: The quality of readiness of the manuscript must be improved.

Response: Thank you for the thorough review done on this work to improve the quality of the manuscript.

Comment: There are some examples of better uses of English in the PDF manuscript

Response: Comments on the annotated PDF manuscript has been attended to and highlighted on the manuscript.

Comment: The introduction can be improved by presenting the critical relevant information to introduce the objectives of this research to the readers.

Response: Significant literature are reviewed and relevant information extracted and discussed through the manuscript.

Comment: The authors should document how they know that the Ni/Mo ratio or concentration was achieved.

Response: The elemental composition of all prepared catalyst obtained from Energy-Dispersive X-ray Spectroscopy (EDXS) is presented in Figure S1 of the supporting information.

Comment: There are other comments across the document. Please address comments or questions accordingly

Response: Thank you. Comments on the manuscript are appropriately addressed.

Reviewer 2 Report

Present work is on the influence of activation techniques to the yield and quality of MWCNT synthesized with corn cob and waste PP.

This is a solid work and should be suitable for the Journal.

i) Some important work such as below seem to be missing, so please do more literature search and improve the references.

Journal of Analytical and Applied Pyrolysis
Volume 94, March 2012, Pages 91-98
Pyrolysis of waste polypropylene for the synthesis of carbon nanotubes

ii) line 427-428 seems to be a rephrase of line 420-421 and can be omitted.

iii)  The magnification of the TEM images are pretty similar. We need lower magnification images on one side (a,c, and e) to better understand the overall morphology and structure.

iv) The yield shown in Table 4 seems unfamiliar. It is rather showing the activity of catalysts.

Can you also derive the ratio of CNT obtained from PP waste(carbon equivalent) ? This should be a more common yield.

Can you measure the amount of NiMo left on each catalyst support? This seems important for the yield of CNT as well.

v) In Fig3, 5, and 7, A and B should be inverted so that it is consistent with other parts. The order should be 0, x, T.

Author Response

Open Review 2

Comment: Some important work such as below seem to be missing, so please do more literature search and improve the references.

Journal of Analytical and Applied Pyrolysis
Volume 94, March 2012, Pages 91-98
Pyrolysis of waste polypropylene for the synthesis of carbon nanotubes

Response: Thank you for the suggested article. More studies from the literature have been presented and discussed through the manuscript.

Comment: line 427-428 seems to be a rephrase of line 420-421 and can be omitted.

Response: Thank you for pointing out this error. The section has been modified. See line 427-429 and line 434-435 as highlighted on the manuscript.

Comment: The magnification of the TEM images are pretty similar. We need lower magnification images on one side (a,c, and e) to better understand the overall morphology and structure.

Response: The TEM micrographs on Figure 6 is been modified. See line 349-350 on the manuscript.

Comment: The yield shown in Table 4 seems unfamiliar. It is rather showing the activity of catalysts. Can you also derive the ratio of CNT obtained from PP waste (carbon equivalent)? This should be a more common yield. Can you measure the amount of NiMo left on each catalyst support? This seems important for the yield of CNT as well.

Response: Thank you for your comments. The formula used for calculating yield was based on catalyst activity (describing catalytic performance in term of different activation techniques employed) as highlighted on the manuscript. See line 193-194.

According to Zhuo and Levendis [1], “there are at least 3 major definitions of production yield: (1.) Mass production rate of CNTs expressed in g/h. (2.) Mass of CNTs over the mass of the catalyst expressed in g/g. (3.) Mass of CNTs over the mass of the feedstock expressed in g/g or %”. Points 1 and 2 are usually used traditionally to express CNTs production processes where the focuses are either on the end product (CNTs) or on the activities of the catalysts, respectively. Point 3 is commonly employed on waste-to-CNTs conversion processes to express their conversion efficiency. Therefore, since the objective of the study was to establish a link between activation and catalytic performance in nanomaterials synthesis (catalyst activity) using waste polymer material as feedstock. Hence, the focus was not on catalyst activity according to point (2) mentioned above.

Reviewer 3 Report

The manuscript by Modekwe et al. reports on the synthesis of carbon nanotubes partially filled with NiMo particles. The work appears to highlight a pivotal role of high temperature pyrolyzed corn cob biomass.  It focuses on the activation methods of the biochar and on the resulting properties of the produced carbon nanotube (CNT) materials. While the topic is interesting, the obtained results lack of novelty and most likely indicate an early stage work. It is not clear the role of the biochar on the nanotube growth.  The produced multiwall CNTs  exhibith a defect-rich arrangement which has been frequently shown in early reports. This is visible also in the Raman spectra, where the authors did not label the D' component, that is indicative of the defective nature of the CNTs.

Author Response

Open Review 3

The manuscript by Modekwe et al. reports on the synthesis of carbon nanotubes partially filled with NiMo particles. The work appears to highlight a pivotal role of high-temperature pyrolyzed corn cob biomass.  It focuses on the activation methods of the biochar and on the resulting properties of the produced carbon nanotube (CNT) materials. While the topic is interesting, the obtained results lack novelty and most likely indicate an early-stage work. It is not clear the role of the biochar on the nanotube growth.  The produced multiwall CNTs exhibit a defect-rich arrangement which has been frequently shown in early reports. This is visible also in the Raman spectra, where the authors did not label the D' component, which is indicative of the defective nature of the CNTs.

Response: Thank you for reviewing this manuscript. Here, in this study, two different waste materials: waste corn cob (char) and waste polypropylene were valorised and utilized as catalyst support and carbonaceous feedstock in the production of value-added materials, respectively.

The novelty in this study was that no study in open literature has explored the use of char obtained from corn cob (waste biomass) as catalyst support in the synthesis of CNTs secondly, for the first time in the open literature, the effect of different char activation techniques on the morphology, quality, and yield of CNTs produced from waste plastics was reported.

 The defective nature of as-produced CNTs was shown in Table 4 on lines 394 and line 358 -359 on the manuscript.

Reviewer 4 Report

This is an interesting paper. This reviewer has the following comments/suggestions

  1. There are a number of methods of producing carbon nanomaterials, nanotubes, fibres using natural products that are relevant to this research and would be useful to readers new to this field. More examples could be provided by the authors. Further examples include the following

- Interconnected carbon nanosheets derived from hemp for ultrafast supercapacitors with high energy, H Wang, Z Xu, A Kohandehghan, Z Li, K Cui, X Tan, TJ Stephenson et al. ACS nano 7 (6), 5131-5141, 2013.

- Carbon nanofibres from fructose using a light-driven high-temperature spinning disc processor, H Lu, RA Boulos, BCY Chan, CT Gibson, X Wang, CL Raston, HT Chua, Chemical Communications 50 (12), 1478-1480, 2014.

Please note it is not compulsory for the authors to cite the above articles. It is only a suggestion.

  1. The authors provide examples of applications of carbon nanotubes but only provide 3 references. Please expand these references and consider other useful applications such as, for example, energy storage (batteries) and scanning probe microscopy.

- Carbon nanotubes for lithium ion batteries, BJ Landi, MJ Ganter, CD Cress, RA DiLeo, RP Raffaelle, Energy & Environmental Science 2 (6), 638-654, 2009.

- Solution based methods for the fabrication of carbon nanotube modified atomic force microscopy probes, AD Slattery, CJ Shearer, JG Shapter, JS Quinton, CT Gibson, Nanomaterials 7 (11), 346, 2017.

Please note it is not compulsory for the authors to cite the above articles. It is only a suggestion.

  1. There are also other sources for producing activated carbon or char for a host of applications. Novel sulfur based polymers are a relatively new type of material that have been used for applications such as mercury, oil spill and iron capture but have also been repurposed as activated carbon and have continued to demonstrate excellent mercury capture response. Sulfur polymers are typically created from waste products such as elemental sulfur, which there are huge amounts, and limonene or canola oil. A paper that highlights the activated carbon application is below

 -  Carbonisation of a polymer made from sulfur and canola oil, M Mann, X Luo, AD Tikoalu, CT Gibson, Y Yin, R Al-Attabi, GG Andersson et al. Chemical Communications 57 (51), 6296-6299, 2021.

Please note it is not compulsory for the authors to cite the above article. It is only a suggestion.

  1. What is the current cost of multi-walled nanotubes? The authors mention they are expensive. How expensive?

  1. How environmentally friendly or green is the proposed method of producing carbon nanotubes?

  1. More details need to be provided for all characterisation techniques. For example for the Raman data the following information needs to be provided

- What was the spectral grating used

- what was the approximate laser power?

- What was the integration time for the spectra acquired?

- How was the spectra calibrated (did you use silicon for example)?

- How many spectra were acquired to calculate the quite Ig/Id ratios. At least 10 should have been taken and an average calculated. What was the error on these values?

Author Response

Open Review 4

This is an interesting paper. This reviewer has the following comments/suggestions

 Comment: There are a number of methods of producing carbon nanomaterials, nanotubes, fibres using natural products that are relevant to this research and would be useful to readers new to this field. More examples could be provided by the authors. Further examples include the following

- Interconnected carbon nanosheets derived from hemp for ultrafast supercapacitors with high energy, H Wang, Z Xu, A Kohandehghan, Z Li, K Cui, X Tan, TJ Stephenson et al. ACS nano 7 (6), 5131-5141, 2013.

- Carbon nanofibres from fructose using a light-driven high-temperature spinning disc processor, H Lu, RA Boulos, BCY Chan, CT Gibson, X Wang, CL Raston, HT Chua, Chemical Communications 50 (12), 1478-1480, 2014.

 Please note it is not compulsory for the authors to cite the above articles. It is only a suggestion.

Response: Thank you for the suggested articles. Other substrates utilized in the synthesis of carbon nanotubes are mentioned in the manuscript. See line 41-43 and 47-48 on the manuscript.

Comment: The authors provide examples of applications of carbon nanotubes but only provide 3 references. Please expand these references and consider other useful applications such as, for example, energy storage (batteries) and scanning probe microscopy.

- Carbon nanotubes for lithium ion batteries, BJ Landi, MJ Ganter, CD Cress, RA DiLeo, RP Raffaelle, Energy & Environmental Science 2 (6), 638-654, 2009.

- Solution based methods for the fabrication of carbon nanotube modified atomic force microscopy probes, AD Slattery, CJ Shearer, JG Shapter, JS Quinton, CT Gibson, Nanomaterials 7 (11), 346, 2017.

Please note it is not compulsory for the authors to cite the above articles. It is only a suggestion.

Response: Thank you for the suggested articles. Some other applications of CNTs have been highlighted and cited accordingly in the manuscript. See line 38-41.

Comment: There are also other sources for producing activated carbon or char for a host of applications. Novel sulfur based polymers are a relatively new type of material that have been used for applications such as mercury, oil spill and iron capture but have also been repurposed as activated carbon and have continued to demonstrate excellent mercury capture response. Sulfur polymers are typically created from waste products such as elemental sulfur, which there are huge amounts, and limonene or canola oil. A paper that highlights the activated carbon application is below

 -  Carbonisation of a polymer made from sulfur and canola oil, M Mann, X Luo, AD Tikoalu, CT Gibson, Y Yin, R Al-Attabi, GG Andersson et al. Chemical Communications 57 (51), 6296-6299, 2021.

Please note it is not compulsory for the authors to cite the above article. It is only a suggestion.

 Response: Thank you for the suggested articles. However, the focus is not on the sources of activated carbon or other applications but on the effect of char activation methods on CNTs quality, yield and morphology and its application as support catalyst in the synthesis of CNTs. Also, corn cob was chosen as the source of char because of its abundance in South Africa as one of the major agricultural wastes.

Comment: What is the current cost of multi-walled nanotubes? The authors mention they are expensive. How expensive?

Response: Prices of SWCNTs were about $2000 kg-1 and MWCNTs $100 kg-1 [2]

Comment: How environmentally friendly or green is the proposed method of producing carbon nanotubes?

Response: The proposed method is environmentally benign. No harmful or noxious gases were released into the environment from this process.  A blend of condensable and non-condensable hydrocarbons were produced within the reactor. Again, all produced gases were used as a carbon source to produce CNMs and exhaust gases were collected via a condenser system. 

Comment: More details need to be provided for all characterisation techniques. For example, for the Raman data, the following information needs to be provided

- What was the spectral grating used

- what was the approximate laser power?

- What was the integration time for the spectra acquired?

- How was the spectra calibrated (did you use silicon for example)?

- How many spectra were acquired to calculate the quite Ig/Id ratios. At least 10 should have been taken and an average calculated. What was the error on these values?

Response: Detailed Raman spectroscopy analysis techniques are provided in the manuscript as highlighted. See Line 213-215.

Round 2

Reviewer 3 Report

The authors have revised the manuscript, however the interpretation of the defective nature of the CNTs is still not acccurate. It is important to include additional deconvolution analyses to clearly detect the type of defects nucleated in the CNT-walls. The D' component is still not identified in the Raman analyses (Fig.7). It appears that possible contributions arising from defective carbon at the K and M point may be present. It is suggested the use of Lorentzian fitting. The manuscript lacks of comparisons with other published works. Additional TEM/HRTEM analyses of the CNT layers are needed to evaluate the quality of the CNTs. The authors need to also comment on the growth-aspects, especially considering the presence of a discontinuous inner CNT-capillary.  Further, the growth-product seem to consist of a mixture of nanotube and onions (CNOs). This is not well addressed in the manuscript. The authors need to comment on CNT-quality and on the possible way of purifying the growth product from the CNOs by-products.

Author Response

Point to Point Response to Reviewer’s comments

Comment: The authors have revised the manuscript, however the interpretation of the defective nature of the CNTs is still not accurate. It is important to include additional deconvolution analyses to clearly detect the type of defects nucleated in the CNT-walls. The D' component is still not identified in the Raman analyses (Fig.7). It appears that possible contributions arising from defective carbon at the K and M point may be present. It is suggested the use of Lorentzian fitting.

Response: Thanks for your comments. Additional deconvoluted Raman spectra of all synthesized CNTs were undertaken using the Lorentzian function are shown in Figure S2 of the supplementary information. See line 374-378 of the manuscript.

Comment: The manuscript lacks of comparisons with other published works.

Response: See line 378 – 381, 467 - 472 and Table 5 (line 426 – 427) of the manuscript.

Comment: Additional TEM/HRTEM analyses of the CNT layers are needed to evaluate the quality of the CNTs.

Response: TEM analysis is shown in line 353 – 354 of the manuscript.

Comment: The authors need to also comment on the growth-aspects, especially considering the presence of a discontinuous inner CNT-capillary. 

Response: The growth model are discussed in line 344 – 347.

Comment: Further, the growth-product seem to consist of a mixture of nanotube and onions (CNOs). This is not well addressed in the manuscript.

Response: Thank you for your comments. There is no formed CNOs in the products. The growth-product consists of more of residual metal catalysts, amorphous carbon and CNTs. As confirmed by the TEM, XRD and EDXS analysis. 

Comment: The authors need to comment on CNT-quality and on the possible way of purifying the growth product from the CNOs by-products.

Response: Thank you. Synthesized materials were cleaned-up using mild acid treatment (0.1M HCl), see line 193 – 194 of the manuscript. However, further strong oxidizing acid treatment which may result in functionalization and modification of the CNTs walls may be undertaken prior to its application.
